# GABA Release from Astrocytes in Health and Disease

**DOI:** 10.3390/ijms232415859

**Published:** 2022-12-13

**Authors:** Werner Kilb, Sergei Kirischuk

**Affiliations:** Institute of Physiology, University Medical Center of the Johannes Gutenberg University, Duesbergweg 6, 55128 Mainz, Germany

**Keywords:** Best1, GAT3, Alzheimer’s disease, Huntington’s disease, epilepsy, GABA synthesis, autism spectrum disorder

## Abstract

Astrocytes are the most abundant glial cells in the central nervous system (CNS) mediating a variety of homeostatic functions, such as spatial K^+^ buffering or neurotransmitter reuptake. In addition, astrocytes are capable of releasing several biologically active substances, including glutamate and GABA. Astrocyte-mediated GABA release has been a matter of debate because the expression level of the main GABA synthesizing enzyme glutamate decarboxylase is quite low in astrocytes, suggesting that low intracellular GABA concentration ([GABA]_i_) might be insufficient to support a non-vesicular GABA release. However, recent studies demonstrated that, at least in some regions of the CNS, [GABA]_i_ in astrocytes might reach several millimoles both under physiological and especially pathophysiological conditions, thereby enabling GABA release from astrocytes via GABA-permeable anion channels and/or via GABA transporters operating in reverse mode. In this review, we summarize experimental data supporting both forms of GABA release from astrocytes in health and disease, paying special attention to possible feedback mechanisms that might govern the fine-tuning of astrocytic GABA release and, in turn, the tonic GABA_A_ receptor-mediated inhibition in the CNS.

## 1. Introduction

Astrocytes are the main glial cells in the central nervous system (CNS). Being non-excitable cells, astrocytes fulfill multiple functions, including extracellular ion homeostasis, neurotransmitter uptake, regulation of synapse number, or controlling local blood flow (for recent review, [1,2]). Astrocytes can sense neuronal activity via a variety of membrane-located ionotropic and metabotropic receptors, including the receptors for neurotransmitters glutamate, GABA or ATP, and thereby actively adapt their functions to the physiological requirements (for review, [3,4,5,6,7]). Receptor activation on astrocytes can induce Ca^2+^ and Na^+^ transients, which frequently propagate via the astrocytic syncytium in the form of waves [8,9].

In addition to their homeostatic uptake functions, astrocytes can release biologically active substances, the so-called gliotransmitters, including glutamate [10], D-serine [11], and ATP [12]. Astrocytes possess various mechanisms, which enable the release of substances into the extracellular space, including vesicular release ([10], for review [13]), release via purinergic P_2X7_ receptors [14] or “hemichannels” [15], via the reversal of the neurotransmitter transporter ([16], for review [17]) and through volume-sensitive anion channels [18]. Originally, all the above pathways have been described for glutamate, the main excitatory neurotransmitter in the brain. However, the results of recent studies demonstrate that GABA, the main inhibitory neurotransmitter in the CNS, can be also released from astrocytes. Similar to neuronal GABA transporters (GATs, for review [19]) astrocyte-located GATs can operate in the reverse mode, i.e., releasing GABA [20,21]. In addition, bestrophins 1 (Best1), originally characterized as a swelling- and Ca^2+^-activated Cl^−^ channel [22], has been demonstrated to be permeable not only for glutamate but also for GABA [23]. Best1 channel was originally identified in the retinal pigment epithelium [24], and its mutations have been demonstrated to underlie several retinopathies (for review, [25]). The Best1-mediated GABA release can potentially affect the excitation-to-inhibition (E/I) balance in the brain. Thereby, mutations in this channel have been attributed to neurodevelopmental disorders, including autism spectrum disorder (ASD) and attention deficit/hyperactivity disorder (ADHS, [26]).

However, previous immunohistochemical data reported low levels of the GABA synthesizing enzyme glutamate decarboxylase (GAD) in astrocytes, suggesting that the intracellular GABA concentration ([GABA]_i_) is too low for non-vesicular release [27]. However, recent data demonstrate that astrocytes in various brain regions are not only capable of synthesizing sufficient amounts of GABA, partially via GAD independent pathways, but can also release GABA (see below). In addition, there is evidence that both processes are strongly modulated under pathological conditions [28,29].

In this review, we summarize the available results about GABA release from astrocytes in health and disease and discuss a possible involvement of several release mechanisms to tune the extracellular GABA concentration ([GABA]_o_) and in turn the tonic GABA_A_ receptor (GABA_A_R)-mediated inhibition.

## 2. GABA Synthesis in Astrocytes

Up to recently, the GABA has been considered to be synthesized mostly by neurons. This conclusion was based on biochemical data, suggesting that the astrocyte expression of GAD, as the classical enzyme synthetizing GABA is too low to produce a physiologically relevant amount of GABA [27,28]. In line with this hypothesis, previous studies revealed that the GABA concentration in cultured astrocytes ranges from 0.2 to 0.9 mM [30]. Only after transfection of an astrocytic cell line (C8S) with isoform 67 of the of the glutamate decarboxylase (GAD67), these transfected astrocytes are capable of synthesizing and even releasing GABA via GATs [31]. However, in addition to the GAD-dependent pathway, GAD-independent pathways for the synthesis of GABA have been described (for recent reviews, [2,6]). Cultured astrocytes are able to synthesize GABA from the polyamine putrescine [32,33], thereby utilizing a GAD-independent pathway of GABA synthesis (Figure 1). Recent studies performed both in brain slices and in vivo confirmed the capability of astrocytes to synthesize GABA in both GAD-dependent and GAD-independent ways, and moreover, that the amount of synthesized GABA depends on physiological/pathophysiological conditions [23,33,34,35,36,37].

Immunohistochemical approaches revealed that both Bergmann glial cells and lamellar astrocytes in the cerebellum contain a detectable amount of GABA under physiological conditions. Using an electrophysiological approach, the minimal [GABA]_i_ required to support the measured level of tonic GABA_A_R-mediated inhibition is estimated to be about 3 mM [23]. GABA synthesis in Bergmann glial cells is driven by monoamine oxidase B (MAO-B) from the polyamine putrescine ([23], Figure 1). Using immunogold labeling technique, the GABA concentration in Bergmann glial cells is estimated to be 5–10 mM [36,38]. In Alzheimer’s disease mouse models, MAO-B-mediated GABA synthesis in reactive astrocytes in the hippocampus is potentiated by about five times as compared to control animals [35,37]. Additionally, it has been demonstrated that thalamic astrocytes synthesize GABA via diamine oxidase (DAO) and aldehyde dehydrogenase 1a1 (Ald1a) ([34], Figure 1). In this case, intra-astrocytic GABA concentration has been estimated to be between 4 and 14 mM [34]. Intriguingly, in hippocampal astrocytes from human Alzheimer patients and from 5xFAD mice, an animal model of Alzheimer disease [39], elevated levels of GAD67 have been reported [28]. It was demonstrated that under these conditions, astrocytes are not only GABA synthesizing cells, but can release GABA, supporting tonic inhibition of granule cells in the dental gyrus [28].

The above-mentioned data demonstrates that the [GABA]_i_ of astrocytes is region dependent and may be as high, at about 10 mM. The extracellular GABA concentration ([GABA]_o_) is region dependent as well and amounts to 0.25 µM in the cortex [40], 0.8 µM in the hippocampus [41] and 3.5 µM in thalamus [34]. Such a strong transmembrane GABA gradient might enable GABA release, both via GABA-permeable channels, following the chemical gradient, or via secondary active GABA transporters, using the electrochemical driving force.

## 3. GABA Release from Astrocytes in Health

### 3.1. Volume-Regulated Anionic Channels—Bestrophin1

As astrocytes express aquaporins [42], their membrane is permeable for water. Osmotic gradients between the extracellular and intracellular solutions drive water fluxes across the cellular membrane, influencing the cell volume in this way. An orchestrated functioning of ionic channels and transporters is required to keep the cell volume unchanged or allow controlled cell swelling or shrinkage (for recent reviews, [43,44]). Several volume-gated ion channels contribute to the volume control of astrocytes. Volume-regulated anionic channels are outwardly rectifying anion channels activated by cell swelling, encoded by the SWELL1 gene, and permeable for larger anionic or uncharged molecules such as glutamate or GABA [45].

Bestrophins are a family of swelling- and Ca^2+^-activated Cl^−^ channels [46]. Four bestrophins (Best1–4) have been identified [47]. Bestrophins were identified in the human genome in relation with the disease Best vitelliform macular dystrophy (BVMD, [48]). More than 200 mutations in *Best1* gene are known to be linked with different retinopathies [25,49]. The Best1 channel is predominantly expressed in the retinal pigment epithelium and is localized to the basolateral plasma membrane [24]. Best1 is characterized as a protein with multiple functions, including a Ca^2+^-activated anion channel, a Ca^2+^ channel modulator and a volume-sensitive anion channel (for review, [47,50]). In addition to Cl^−^ ions, all bestrophins are permeable for HCO_3_^−^ [51] and for relatively large anions, such as aspartate and glutamate (Figure 2A, for recent review; [52,53]). These properties of Best1-channel lead to the hypothesis that distorted regulation of intracellular Ca^2+^ signaling [54,55] and/or pH homeostasis [51] may underlie the Best1 mutation-related retinopathies (for recent review, [25]). Moreover, diabetic retinopathy is accompanied by a reduction in GABA release in the retina, leading to hyperexcitation (for review, [56]), but the role of possible Best1-mediated GABA release in the retina needs further investigation.

In situ hybridization and immunohistochemical data demonstrated a strong Best1 expression in the hippocampus, thalamus and cerebellum, both in neurons and astrocytes [57]. The functional expression of Best1 has been confirmed by electrophysiological experiments in cultured mouse hippocampal astrocytes [57]. In addition, the observation that the application of the hypo-osmic solution stimulated the non-synaptic, astrocyte-mediated release of both GABA and glutamate in the mouse hippocampus, suggests a possible involvement of volume-regulated channels permeable for both glutamate and GABA [58]. Recent studies reported that Best1 is also permeable for GABA, with a GABA permeability of 0.27 as compared to Cl^−^ [59,60], which is lower than that for glutamate (0.47–0.67). However, the GABA permeability of Best1 was estimated electrophysiologically, despite the fact that GABA is a zwitterionic amino acid and mostly uncharged at physiological pH. Thus, it can be assumed that the GABA permeability of Best1 may be much higher. This question definitely needs further investigations.

Lee at al. [23] performed the first detailed study that confirmed the Best1-mediated GABA release from astrocytes in the cerebellum. Best1 is a Ca^2+^-activated anionic channel and its half-maximal activation was determined to be at [Ca^2+^]_i_ of 150 nM. This concentration is quite close to the resting [Ca^2+^]_i_ of astrocytes, suggesting the possibility of tonic GABA release under resting conditions. Pharmacological blockade or genetic silencing of Best1 in cerebellar astrocytes reduces the GABA_A_ receptor (GABA_A_R)-mediated tonic current in cerebellar granule cells by about 75%. These results indicate that Bergmann glia cells and lamellar astrocytes in the mouse cerebellum tonically release GABA via Best1 channels under resting physiological conditions [23]. Subsequent studies by Yoon et al. revealed that Best1 channels are strongly expressed in several thalamic nuclei [61]. Thalamo-cortical projection neurons in the ventrobasal (VB) nucleus of thalamus exhibit sustained tonic GABA_A_R-mediated currents, which remain unchanged after blocking phasic GABA release, indicating a non-vesicular source of extracellular GABA [34]. As [GABA]_o_ in thalamus amounts to 3.5 µM [34], it seems unlikely (see Equation (1) and Figure 2) that the reverse mode of these GABA transporters mediates sufficient GABA release. However, the tonic GABA_A_R-mediated current is reduced to about 30% by the pharmacological blockage of Best1 or selective genetic ablation of Best1 in thalamic astrocytes [34]. Chelating of intracellular Ca^2+^ with BAPTA significantly reduces the tonic GABA_A_R-mediated currents in thalamic neurons, while they are potentiated by an elevation of [Ca^2+^]_i_ in astrocytes, confirming the involvement of the Ca^2+^-modulated Best1 channel. Thus, similar to cerebellar astrocytes, [GABA]_i_ is quite high in thalamic astrocytes and Best1 channels the mediate constitutive [Ca^2+^]_i_ dependent release GABA under resting physiological conditions [34].

In addition to the cerebellum and thalamus, the measurable expression of Best1 channels has been observed in hippocampal astrocytes [57,61]. However, in the hippocampus, the main source of the extracellular GABA is the phasic GABAergic transmission mediated by interneurons [62], and the pyramidal cell do not demonstrate any tonic GABAergic inhibition unless GAT1 is blocked [63]. As only about 20% of hippocampal astrocytes are GABA positive, the GABA release from astrocytes under physiological conditions is not detectable despite the strong Best1 expression in hippocampal astrocytes [61]. Moreover, astrocytic Best1 channels are located in the astrocytic distal processes and positioned closely to excitatory synapses, indicating that probably glutamate, rather than GABA, is released via Best1 channels in the hippocampus under physiological conditions [61,64].

The hypothesis that the spatial distribution of Best1 channels in the astrocytic membrane may depend on [GABA]_i_ has been addressed in a recent study [65]. In control mice, astrocytes in the hippocampal CA1 region containing a low amount of GABA and Best1 channels are mostly located in the proximity of glutamatergic synapses. In the APP/PS1 mouse model of Alzheimer’s disease, the MAO-B activity is potentiated in hippocampal astrocytes, resulting in an elevated [GABA]_i_ [35]. Under these conditions, the spatial Best1 distribution is shifted closer to GABAergic synapses, while in the hippocampus of MAO-B-KO mice (lower [GABA]_i_), the Best1 channels were located closer to glutamatergic synapses [65].

In summary, in brain regions where resting GABA concentration in astrocytes is high, namely in the cerebellum and thalamus, the tonic Best1-mediated GABA release takes place under resting physiological conditions. The amount of GABA released is modulated by intracellular Ca^2+^ activity and the transmembrane GABA gradient. However, the Best1 permeability for GABA, and so the selectivity for GABA as compared to glutamate, remains to be further investigated. Low intracellular GABA concentration probably results in the Best1 channel relocation closer to glutamatergic synaptic contacts and switches the released gliotransmitter to glutamate.

### 3.2. GAT-Mediated GABA Release

In the classical view, GATs are supposed to remove GABA from the extracellular space. GAT-mediated uptake of one GABA molecule is accompanied with the cotransport of one Cl^−^ and two Na^+^ ions [66]. Because under physiological pH GABA is predominantly a zwitterionic molecule, the transport process is electrogenic.

As all three molecules/ions, namely GABA, Na^+^ and Cl^−^, are transported independently from each other, the GAT reversal potential can be estimated using the following thermodynamic equation [17]:(1)$$E_{rev}=-\left[\frac{RT}{\left(2Z_{Na}+Z_{Cl}\right)\times F}\right]\times \ln\left(\frac{{\left[GABA\right]}_{i}}{{\left[GABA\right]}_{o}}\times \frac{{\left[Na\right]}_{i}^{2}}{{\left[Na\right]}_{o}^{2}}\times \frac{{\left[Cl\right]}_{i}}{{\left[Cl\right]}_{o}}\right)$$

Importantly, the GAT-mediated transport process can be switched into the opposite direction, the reverse mode, thereby releasing GABA. GAT reversal occurs if the membrane potential is positive to the GAT reversal potential [17], which can be caused by a membrane depolarization or if the intracellular Na^+^ concentration ([Na^+^]_i_) increases high enough (Figure 2B). Indeed, the depolarization-induced GAT-mediated GABA release was reported from retinal horizontal cells, growth cones of neurons and various neurons in cultures or acute brain slices [19,67,68,69]. In addition, tonic GABA_A_R-mediated inhibition was reported to be mediated by the GAT3-mediated GABA release in the acute slice preparation of the neocortex [20]. In the marginal zone of the early postnatal cortex, the GAT3-mediated GABA release was demonstrated to result in a presynaptic GABA_B_R-mediated suppression of GABAergic transmission [70]. Interestingly, GAT1 is reported to be expressed both in neurons and astrocytes [71], while GAT3 is predominantly located to astroglial processes [72], suggesting that GAT-3, in particular, contributes to the tonic GABA release from astrocytes in the neocortex. In the hippocampus, however, astrocytes are capable of releasing GABA only after a strong elevation of [Na^+^]_i_, as a result of facilitating the EAAT-mediated glutamate uptake, occurring, for example, during epileptic-like activity [21,73].

GAT reversal potential in astrocytes can be estimated from Equation (1) and values available in the literature ([Na^+^]_o_ = 140 mM, [Cl^−^]_i_ = 135 mM, [Na^+^]_i_ = 15 mM [74,75,76,77], [Cl^−^]_i_ = 30–40 mM [78,79]. The extracellular GABA concentration ([GABA]_o_) strongly depends on the brain region. In the early postnatal cerebral cortex, [GABA]_o_ has been reported to be 0.25 µM [40], in the adult hippocampus ~0.8 µM [41] and in the thalamus ~3.5 µM [34]. Using these numbers, one can estimate the [GABA]_i_ required to set the GAT reversal potential close to the resting potential of astrocytes (ca. −80 mV), obtaining a [GABA]_i_ of ~1 mM in the cerebral cortex, ~4 mM in the hippocampus, and ~10 mM in the thalamus (Figure 2C). These estimates demonstrate that GAT reversal under resting conditions is more probable in the cerebral cortex [20,76]. Moreover, the relative position of the reversal potential, and thus the direction and strength of the astrocytic tonic GABA release can be modulated by [Na^+^]_i_ transients, occurring for example via the activation of EAATs (Figure 3A [80]).

In the hippocampus, the main source of extracellular GABA is supposed to have a neuronal origin, and GAT1 operating in the uptake mode is the dominant GAT isoform [41,82]. However, further studies revealed that astrocytic GAT3 contributes to setting the GABA levels in the extracellular space [83]. However, under physiological conditions, the astrocytic [GABA]_i_ appears to be relatively low, and the GAT3 operate in the uptake mode, reducing extracellular [GABA]_o_. This suggests that in the hippocampus astrocytic, [GABA]_i_ is similar to that in the cerebral cortex, i.e., about 1 mM [73], and [GABA]_o_ amounts to 0.8 µM [41,73]; an elevation of [Na^+^]_i_ to about 25–30 mM is required to result in GAT reversal (Figure 2D). Because EAATs cotransport three Na^+^ ions with one glutamate molecule, EAAT-mediated glutamate uptake strongly affects [Na^+^]_i_ [76]. Consequently, elevated neuronal activity, for example epileptic-like activity, could potentially reverse GAT3. Indeed, the activation of EAATs with the EAAT substrates glutamate or aspartate leads to the GABA release from astrocytes [21]. Interestingly, the GAT1-mediated GABA uptake does not reverse during epileptic-like activity [84]. As GAT1 in the hippocampus is located predominantly to neurons, this result can be explained by a lower resting [Na^+^]_i_, of about 8 mM in neurons [85]. Thus, because of the relatively high [Na^+^]_i_ in glial cells, the astrocytic GAT3 may fulfill a specific protecting role under pathophysiological conditions. Indeed, using a low [Mg^2+^]_o_ in vitro model of epilepsy, a GAT3-mediated GABA release has been demonstrated to reduce the duration of seizure-like events. In addition, the blockade of GAT3-mediated GABA release in vivo reduces the power of the gamma-range oscillation, indicating that the GAT3-mediated GABA release is functional under physiological conditions [73].

Similar to the hippocampus, GAT1 is the dominant GABA transporter in the striatum, a brain region in which extracellular GABA tunes the activity of output neurons [86] and dopamine release [87]. Inhibition of GAT1-mediated GABA uptake discloses an additional GAT3-mediated removal of GABA in the striatum [88]. The physiological role of GAT3-mediated release from astrocytes was investigated in the recent work of Yu at al. [81]. This group reduced intracellular Ca^2+^ activity selectively in striatal astrocytes by viral overexpression of the plasma membrane Ca^2+^ ATPase type 2 (PLMCA2). Interestingly, the dumping of astrocytic [Ca^2+^]_i_ transients affected mouse behavior, as evident by potentiated self-grooming. Electrical activity of medium spiny neurons (MSNs), striatal output neurons, was reduced, although neither glutamatergic nor GABAergic phasic synaptic transmission has been modified. The observed reduction in MSN activity was mediated by tonic GABAergic inhibition via extrasynaptic GABA_A_ receptors. Unexpectedly, the blockade of GAT3, but not of GAT1, rescued the strength of tonic GABA_A_R-mediated inhibition and alleviated the self-grooming behavior. The further experiments demonstrated that the reduction in Ca^2+^ activity reduced the number of GAT3 inserted in the plasma membrane, while the mere GAT3 expression was unaffected [81]. Thus Ca^2+^-dependent GAT3 trafficking determines the extracellular GABA concentration. This work demonstrates that astrocytes may control tonic GABA_A_R-mediated inhibition not necessarily via GABA release but also via the tunable GABA uptake.

In summary, tonic GAT3-mediated GABA release from astrocytes occurs in the cerebral cortex probably because of low [GABA]_o_. In case of higher extracellular GABA concentration, such as in the hippocampus, additional elevation of [Na^+^]_i_ is required to change the direction of GAT3-mediated GABA transport. Furthermore, [Ca^2+^]_i_ influences the number of plasma membrane located in GAT3 and in turn the strength of the GAT3-mediated release/uptake.

## 4. GABA Release from Astrocytes in Disease

### 4.1. Alzheimer’s Disease

Alzheimer´s disease (AD) is a neurodegenerative disorder with an increasing prevalence in aging populations. Several lines of evidence suggest that amyloid β (Aβ) peptide and tau protein aggregations trigger a cascade of pathological changes, including the weakening of synaptic transmission and synapse loss (for review, [89]). Astrocytes as a part of the tripartite synapse are involved in the development of AD pathology (for recent review, [90]).

Elevated tonic GABA_A_R-mediated inhibition in the dental gyrus in the hippocampus has been reported in the 5xFAD mouse model of AD [28]. Immunochemical data demonstrate increased GABA and glutamate levels in GFAP-positive reactive astrocytes. The elevated GABA concentration in astrocytes is accompanied by an increased level of GAD67. These changes are observed in 6- to 8-months old animals after the appearance of Aβ plaques, but not at younger ages (3- to 4 months). A similar elevation of GABA level in reactive astrocytes was detected in human AD patient brains [28]. In addition, immunostaining experiments revealed an increased expression of GAT3 both in mouse reactive astrocytes and human AD brain samples, suggesting that GAT3-mediated GABA release underlies the elevated tonic inhibition in these animals [28]. Indeed, the selective GAT3 blockade significantly reduces the tonic GABA current in granule cells and also rescues the synaptic plasticity and memory deficit in this mouse model [28]. In contrast, the blockade of GAT1 enhances tonic GABA_A_R-mediated currents, confirming that GAT1 in the AD hippocampus still operates in uptake mode [83]. Thus, in the hippocampus of the 5xFAD mouse model of AD, both GABA synthesis and release via GAT3 are specifically facilitated in reactive astrocytes, but not in neurons [28]. Similar results were obtained in another model of AD, the APP^NL−F/NL−F^ knock-in mouse model [91]. Moreover, in this case, both GABA content and GAD67 expression are significantly increased in reactive astrocytes, leading to the GAT3-mediated GABA release and increased tonic inhibition of principal neurons in the CA1 and DG regions of the hippocampus [91]. Unfortunately, the expression of Best1 channel has not been inspected in these publications.

GABA release from reactive astrocytes has also been reported in the hippocampus of APP/PS1 mice, another mouse model of AD [35]. Extracellular GABA, but not glutamate levels are significantly increased in the APP/PS1 mice as compared with WT mice, leading to elevated tonic GABA_A_R-mediated currents. Immunostainings with antibodies against GABA demonstrated that the GABA content, as estimated from the intensity of immunoreactivity, in reactive astrocytes is comparable with that in GABAergic interneurons. In contrast to the results from the 5xFAD AD mouse model [28], the GAD activity is not potentiated in these APP/PS1 mice. Pharmacological tools revealed that the increased level of GABA in astrocytes results from the elevated MAO-B activity, confirming that GABA is synthesized from putrescine (Figure 1) in this case. Moreover, in contrast to the 5xFAD AD mouse [28], GAT1 and GAT3 expressions are not significantly altered in the APP/PS1 mice. The GABA release from reactive astrocytes has been demonstrated to occur via the Best1 channels [35]. Moreover, the Best1 membrane distribution is strongly changed, demonstrating in reactive astrocytes of APP/PS1 mice a major localization in soma and proximal dendrites, instead of the distal microdomain-like distribution observed in WT animals [35]. The GABA release from astrocytes in the hippocampus of APP/PS1 mice has been demonstrated to activate GABA_B_ receptors at GABAergic interneurons, leading to the disinhibition of granule cells [92]. Interestingly, Jo at al. [35] reported that the *Best1* silencing by means of shRNA blocks GABA release in the 5xFAD AD mouse model as well, which also suggests the Best1-mediated GABA release in this AD mouse model. These results contradict the GAT3-mediated release reported by Wu et al. and Aldabbagh et al. [28,91]. Unfortunately, this contradiction was not discussed in this publication. As the GABA elevation in reactive astrocytes and accumulation of Aβ plagues demonstrate a bell-shaped relationship [93], one may suggest that both the level of astrocytic GABA and active GABA release mechanism depend on the disease stage, similar to the data reported for epilepsy (see below, [94]).

### 4.2. Huntington’s Disease

Huntington´s disease (HD) is a neurodegenerative disorder triggered by poly-glutamine expansion in the huntington protein [95]. Previous studies reported decreased levels of EAATs, the astrocytic glutamate transporters [96,97,98], and suggested elevated extracellular concentration of glutamate [99] and excitotoxic cell death as a possible cause of HD symptoms´ manifestation. Parallel to glutamatergic system malfunctions, GABAergic signaling, including tonic GABA_A_R-mediated inhibition, is demonstrated to be distorted as well (for recent review, [100]). In the mouse striatum tonic, the GABA_A_R-mediated currents in striatal output neurons (SONs) under resting conditions were detected from P19 on. The blocking of GAT1, but not GAT3, discloses tonic currents at younger ages and potentiated them at older ones, indicating that GAT1 is the dominant GABA uptake mechanism in the striatum and operates in the uptake mode [86]. When GAT1 is pharmacologically blocked, the GAT3 also operates in the uptake mode [88]. However, as GAT1 is the main GABA uptake mechanism, its blockade would result in an elevation of [GABA]_o_, potentially affecting the direction of the GAT3-mediated GABA transport and thereby obscuring a putative release mode under these experimental conditions (Figure 2C).

Further experiments revealed that in WT animals about half of the striatal extracellular GABA is released synaptically, while the remaining half represents the GABA released via GAT3 from astrocytes [101]. In two mouse models of HD, namely Z-Q175-KI and R2/6, however, both the GAT3-mediated GABA release and tonic GABA_A_R-mediated currents in SONs have been strongly reduced. The increase in [Na^+^]_i_ by the D-aspartate-triggered stimulation of EAAT switched the GAT3 in the release mode, similar to that in WT animals. Thus, it was suggested that in the healthy striatum, the EAAT-mediated glutamate uptake keeps GAT3 in the reverse mode by means of elevated [Na^+^]_i_ (Figure 2D); meanwhile, in HD astrocytes, the reduced glutamate uptake and lower [Na^+^]_i_ fail to reverse GAT3, and it operates in the uptake mode, leading to the hyperexcitability of SONs [101].

Using the R2/6 mouse model of HD, Jiang et al. [102] reported the reduced spontaneous, but potentiated evoked Ca^2+^ signaling in striatal astrocytes. The reduced expression of EAAT2 and the K^+^ channel Kir4.1 has been observed in this model, leading to the elevated extracellular glutamate levels and activation of metabotropic GluRs. Interestingly, these R2/6 mice demonstrate increased self-grooming behavior [103], comparable to the behavior observed after attenuating the tonic GABA_A_R-mediated inhibition in the striatum (by the enhanced Ca^2+^-dependent GAT3-mediated astrocytic GABA uptake upon PLMCA2 overexpression) [81]. Ca^2+^ signaling in striatal astrocytes was reduced in this HD mouse model [103], leading to reduced extracellular GABA and reduced tonic GABA_A_R-mediated inhibition. Thus, it is possible that the facilitated GAT3-mediated GABA uptake underlies the observed changes, similar to the results obtained in the WT mice [81].

The source of astrocytic GABA has been investigated by Yoon et al. [38]. They show that astrocytes in the striatum are capable of synthesizing GABA from putrescine via MAO-B pathway (Figure 1). In line with Wojtowicz et al. [101], astrocytes release GABA under resting physiological conditions. Although the Best1-mediated GABA release from astrocytes has been suggested by Yoon et al., the authors did not provide any data that would allow one to distinguish between GAT3- and Best1-mediated release mechanisms [38].

In summary, the experimental data demonstrate that the GABA release from striatal astrocytes is reduced in HD. However, the question whether the observed reduction results from (1) reduced [Na^+^]_i_ and switch of GAT3 in uptake mode; (2) reduced number of membrane-located GAT3; or (3) reduced [Ca^2+^]_i_ with decreased Ca^2+^-dependent Best1 conductance, remains elusive.

### 4.3. Autism Spectrum Disorder (ASD)

ASD is a neurodevelopmental disorder characterized by repetitive behaviors, communication deficits and disturbed sensory perception. The etiology of ASD remains elusive. The main working hypothesis is that the excitation/inhibition (E/I) imbalance leads to the manifestation of ASD symptoms (for recent review, [104,105]). An elevated E/I ratio might result from changes in glutamatergic and/or GABAergic signaling. Indeed, the selective stimulation of excitatory neurons in the mouse medial prefrontal cortex in vivo is sufficient to induce social behavior deficits, while the selective activation of inhibitory interneurons rescues the observed impairments [106]. In addition to the potentiated glutamatergic synaptic inputs, elevated levels of glutamate in the extracellular space may also lead to excitotoxicity in ASD [107]. Extracellular glutamate concentration is set by the activity of EAATs. Astrocyte-located EAAT2 plays a critical role in controlling extracellular glutamate level and its genetic silencing results in epileptic seizures and premature death [108]. The conditional deletion of EAAT2 in astrocytes reduced its expression in the cortex and striatum by 60–80% [109]. Such a reduction in astrocytic EAAT2 failed to affect either the kinetics of cortico-striatal EPSCs or the paired-pulse ratio of EPSCs, but significantly attenuated the reduction in the EPSC amplitude during repetitive stimulation. These results indicate that the reduced uptake of glutamate by astrocytes potentiates the glutamatergic drive, i.e., shifts the E/I balance to excitation, leading in turn to self-grooming (ASD-like) behavior [109].

Similar to the excitatory glutamate action, the reduction in the extracellular GABA level may shift the E/I ratio towards excitation. Diminishing the Ca^2+^ signaling by the overexpression of Ca^2+^ extruding ATPase in hippocampal astrocytes results in repetitive behavior, typical for ASD. Further analysis revealed no change in EAAT2 expression and a strong overexpression of GAT3, the astrocytic GABA transporter [81]. As GAT3 in the hippocampus under control conditions operates in the uptake mode [83], the GAT3 over-activity reduces the extracellular GABA level and shifts the E/I balance towards excitation. As astrocytic EAAT activity may reverse the Na^+^/Ca^2+^ exchanger via the [Na^+^]_i_ increase [110], intra-astrocytic sodium and calcium signaling may set the EAAT/GAT expression balance and E/I ratio. Interestingly, the E/I imbalance resulting from a distorted balance of astrocytic GAT3 and EAAT2 seems to also underlie the obsessive compulsive disorders (for recent review, [111]).

### 4.4. Epilepsy

Epilepsy is a disorder characterized by excessive network activity. The shift of excitation-to-inhibition (E/I) balance towards excitation is generally suggested to underlie the occurrence of spontaneous network discharges, which manifest as seizures (for review [112,113]). The pharmacological enhancement of GABAergic (inhibitory) transmission may be beneficial to suppress hyperexcitability and prevent epileptic seizures [114,115,116]. In this context, the GABA release from astrocytes and resulting tonic GABA_A_R-mediated inhibition might be a potent tool to rescue the distorted E/I balance [117].

As GABA concentration in hippocampal astrocytes under physiological conditions is relatively low, GATs operate in uptake mode under the control condition [84]. However, the EAAT blockade in vitro and in vivo results in an elevation in the extracellular GABA concentration [21,73]. The observed GABA release is mediated by astrocytic GAT3 [21], leading to the conclusion that the EAAT-induced [Na^+^]_i_ elevation can switch the GAT3 into reverse mode (Figure 2D). Such increased [Na^+^]_i_ may occur under epileptic conditions. Indeed, Heja et al. [21,73] have demonstrated that hippocampal astrocytes are capable of both GABA synthesis from putrescine (Figure 1) and EAAT-dependent GAT3-mediated GABA release. In the low-Mg^2+^ model of epilepsy, the GAT3-mediated GABA release significantly shortens seizure-like events, confirming the inhibitory role of GABA release from astrocytes [21,73]. Similar results were obtained in vivo in a rat model of absence epilepsy, the WAG/Rij model [118,119]. Moreover, these in vivo studies also revealed that the astrocytic polyamine metabolism may modulate the neuronal activity in several ways, including the GAT3-mediated GABA release [118,119].

The strong reduction in the GABAergic interneuron number has been demonstrated in the human and different mouse model of temporal lobe epilepsy [29,120,121,122]. GABAergic interneurons represent the main source of extracellular GABA in the healthy hippocampus [62], and thus the decreased interneuron number should in turn weaken the phasic GABA release. However, unexpectedly, the tonic GABA_A_R-mediated inhibition was elevated already at day 5 after epilepsy induction in the intracortical kainate mouse model [29]. The [GABA]_i_ in hippocampal astrocytes increased from day 5 on and became more pronounced at day 14 and 28. The facilitated GABA synthesis in reactive astrocytes was demonstrated to be mediated by both MAO-B and GAD [29]. Interestingly, the potentiation of MAO-B and GAD activity in astrocytes has been observed also contralaterally; however, without any change in the strength of tonic inhibition on this side. In contrast to the results obtained in the WAG/Rij rat model [118,119], GAT3 did not mediate the facilitated GABA release from the astrocytes in this model of epilepsy [29].

While the mechanism of GABA release was not analyzed by Müller et al. [29], a recent study by Pandit et al. addressed the mechanism of GABA release from astrocytes in the mouse hippocampus in the kainic acid (KA) and pilocarpine (PI) models of epilepsy [94]. In both models on the third day after KA or PI injection, the hippocampal astrocytes demonstrate an increased expression of GFAP, a sign of reactive astrocytes, and an elevated [GABA]_i_. Interestingly, [GABA]_i_ returns to control level at day 7 after KA/PI injection [94], in contrast to the elevated astrocytic [GABA]_i_ at day 14 reported in [29]. Similar to [GABA]_i_, Best1 expression is strongly increased at day 3 and returns almost to the control level thereafter [94]. Phasic GABAergic synaptic transmission is not affected, while the tonic GABA_A_R-mediated currents in CA1 pyramidal neurons are strongly increased. The pharmacological blockade of Best1 channels reduces the strength of the tonic GABA_A_R-mediated inhibition to about 30%. Moreover, the selective expression of Best1 channels in hippocampal astrocytes in Best1-KO mice essentially suppresses seizure susceptibility, indicating an essential role of the Best1-mediated GABA release under epileptic conditions [94]. GAT1 was demonstrated to operate in the uptake mode. Unfortunately, nipecotic acid, a non-selective GAT antagonist, was used to exclude the GAT3-mediated GABA release. As the GAT1 blockade would definitely increase [GABA]_o_ and potentially switch GAT3 in the uptake mode (Figure 2C), a possible role of the GAT3-mediated release for the increased tonic GABA_A_R-mediated inhibition requires further investigations.

In summary, the epileptic activity stimulates (1) GABA synthesis in hippocampal astrocytes and (2) results in astrocytic GABA release. The question of whether the GABA release is driven by the EAAT-mediated [Na^+^]_i_ increase and GAT3 reversal or whether it occurs via the Best1-mediated GABA release requires further experiments.

## 5. Summary: Is GABA Release from Astrocytes Mediated by Best1- or GAT3 or Both?

Theoretical assumptions indicate that, depending on the region and functional state, both the channel- and/or transporter-mediated GABA release is possible, given an astrocytic [GABA]_I_ in the range of 0.5–4 mM and [GABA]_o_ in the range of 0.5–1 µM. For both channel- and/or transporter-mediated GABA releases, experimental evidence has been provided, as summarized in the previous paragraphs and in Table 1. However, to be functionally relevant, the astrocytic GABA release has to be controlled by physiological demands. The physiologically required amount of extracellular GABA has to be adjusted to the detected/measured levels of neuronal activity and, accordingly, astrocytic GABA release/uptake has to be adapted. Below, we suggest some possible theoretical feedback mechanisms.

### 5.1. GAT3-Mediated Release

The GAT3-mediated GABA release is by definition equipped with a negative feedback mechanism. If the GAT3 operates in the release mode, the elevation of [GABA]_o_ shifts its reversal towards more positive values, thus favoring the uptake mode (Figure 2C). In addition to the transmembrane GABA gradient, [Na^+^]_i_, and, to a lesser extent, [Cl^−^]_i_, determines the reversal potential and, in turn, the distinctive extracellular GABA level. The local astrocytic [Na^+^]_i_ depends on the activity of several channels and secondary active transporters, and therefore reflects the neuronal activity levels (for reviews, [123,124,125]). Na^+^-coupled mechanisms, especially EAATs [21], may shift the GAT reversal potential and in turn [GABA]_o_ (Figure 3A). Similar to this process, the EAAT activity may reverse the Na^+^/Ca^2+^ exchanger via the [Na^+^]_i_ increase [126]. The resulting intracellular Ca^2+^ transients may not only trigger the release of other gliotransmitters, they influence GAT3 trafficking, modulating the number of membrane-located GAT3 and in turn the amount of GABA released or taken up [81]. Thus, the strength of the GAT3-mediated GABA release depends not only on the transmembrane GABA gradient but also, via changes in intracellular ionic concentrations on the neuronal activity levels, enabling a (patho-)physiologic condition dependent on the fine-tuning of the GABA release.

### 5.2. Best1-Mediated Release

Best1 is a Ca^2+^-activated anionic channel. This enables the possibility that local [Ca^2+^]_i_ levels determine both the threshold and rate of tonic GABA release via this pathway. Interestingly, Bergmann glia cells in the cerebellum demonstrate variable resting [Ca^2+^]_i_, from about 50 to 250 nM, and the amplitude of somatic ATP-induced Ca^2+^ responses is negatively correlated to the resting somatic [Ca^2+^]_i_ [127]. Similar results were obtained in hippocampal and cortical astrocytes in vitro and in vivo using two-photon microscopy [128]. Moreover, even within a single astrocyte, [Ca^2+^]_i_ is spatially heterogeneous, ranging from 40 to 120 nM, and local [Ca^2+^]_i_ controls the amplitudes of sensory-induced Ca^2+^ transients in vivo [128]. Because a half-maximal activation of Best1 channels occurs at a [Ca^2+^]_i_ of about 150 nM [23], the reported levels of [Ca^2+^]_i_ ranging from about 50 to 250 nM are within the dynamic range of Best1, thus supporting the activity-dependent modulation of the Best1-mediated GABA release. However, mechanisms underlying spatially restricted [Ca^2+^]_i_ levels and thus the focal control of the Best1-mediated GABA release needs further investigations.

If the Best1-mediated release is tonic, such as what is reported in the cerebellum and thalamus [23,34], then all GATs probably operate in the uptake mode, thereby setting [GABA]_o_. The reversal of GAT directionality in Bergmann glial cells was reported to occur when [GABA]_i_ reaches about 10 mM [129]. Using sniffer HEK293T cells expressing high-affinity GABA_C_ receptors, a detectable Best1-mediated GABA release was already observed at the [GABA]_i_ of 3 mM [23], suggesting that the Best1-mediated release probably dominates in the cerebellum. In turn, the elevated [GABA]_o_ will shift the GAT reversal potential to more positive values. Thus, the Best1-mediated release and GAT3-mediated uptake operate in concert to determine [GABA]_o_ (Figure 3B). Interesting, the pharmacologic or genetic blockade of Best1 did not completely eliminate the tonic GABA_A_R-mediated current but reduced it to about 30%, both in the cerebellum and thalamus [23,34]. Definitely, one cannot exclude the possibility that another channel, for example Best2 channel [47], contributes to GABA release, but an alternative explanation may be that a Best1 blockade results in lower [GABA]_o_, which in turn switches GAT3 into the release mode (Figure 3B). The observation that the non-selective GAT blockade with nipecotic acid induces larger tonic currents in the WT thalamus as compared to Best1-KO mice [34] indirectly supports this hypothesis.

### 5.3. Best1-/GAT3-Mediated Release

Interestingly, even if GAT3 operates in the uptake mode, it can indirectly suppress glutamatergic activity. In hippocampal slices, the GAT3-mediated GABA uptake leads to an elevation of [Na^+^]_i_, which in turn switches the Na^+^-Ca^2+^ exchange (NCE) into the reverse mode. The NCE-mediated Ca^2+^ influx can stimulate the ATP release from astrocytes, leading to an adenosine-mediated inhibition of glutamate activity [126]. As the EAAT-mediated glutamate uptake elevates an [Na^+^]_i_ stronger than GATs, one can suggest that the GAT3- and/or EAAT-mediated [Na^+^]_i_ increase reverse NCE, resulting in a near-membrane [Ca^2+^]_i_ increase. Such local Ca^2+^ transients may open Best1 channels, leading to the GABA release (Figure 3C). Therefore, both GABA release pathways may be functional in parallel. Changes in intracellular ionic and GABA concentrations can modulate and/or inactivate one of them, potentiating in turn the remaining one.

### 5.4. Neuroglial Interaction and GABA Release from Astrocytes

Definitely not only changes in astrocytic activity or intracellular homeostasis determine the strength and direction of the GABA transport. In the hippocampus under resting conditions, the extracellular GABA has a mostly neuronal origin [41], and the high level [GABA]_o_ keep GAT3 running in the uptake mode. Strongly elevated neuronal activity, such as during epileptic seizures, may increase the extracellular glutamate concentration and switch the GAT3 into release mode [21]. Physiological neuronal activity may result in the activation of astrocytic metabotropic receptors, including those for glutamate, ATP and GABA, resulting in local or global (waves) Ca^2+^ transients (for recent reviews, [130,131]. Such neuron activity-induced changes in [Ca^2+^]_i_ can open Best1 channels, leading to a direct GABA release, or stimulate the GAT3 internalization, resulting in the reduced GABA uptake and indirect elevation in extracellular GABA level. Therefore, Best1 channels and GAT3 represent possible mechanisms of GABA release from astrocytes. Their (patho-) physiological contribution is determined as mainly neuroglial interactions at several levels.

## 6. Conclusions and Perspectives

In this review, we have summarized the experimental results reporting the GABA release under physiological and pathophysiological conditions. Increasing evidence demonstrates that astrocytes are able to synthesize GABA using different metabolic pathways. Intracellular GABA concentration may be as high as 10 mM. Given that the extracellular GABA concentration is in a low micromolar range, the transmembrane GABA gradient enables both Best1-channel- and GAT3-transporter-mediated GABA release. Indeed, the tonic GABA release from astrocytes has been demonstrated in the cortex, cerebellum, thalamus and striatum. Interestingly, in the cerebellum and thalamus, GABA is predominantly released via the Best1 channel, while in the cortex and striatum, the GAT3-mediated GABA release dominates. In diseases, the GABA release from astrocytes is typically facilitated and again, in some cases, the Best1-mediated release is potentiated, whereas, in other cases, the GAT3-mediated pathway plays the dominating role. The question why nature uses one or the other pathway remains elusive. The expression and trafficking of Best1 and GAT3 is modulated by several intracellular factors, including the intracellular GABA and Ca^2+^ concentration. In addition, the local intracellular Ca^2+^ concentration determines the Best1 conductance, while the intracellular Na^+^ concentration influences the GAT3 reversal potential. We suggest that the astrocytes possess both GABA release pathways. Local ionic signaling and GABA gradients, also as a consequence of neuronal activity, may play a decisive role to switch on/off a particular mechanism of GABA release. Elucidating the mechanisms governing GABA release may help to develop therapeutic tools for the treatments of pathophysiological conditions.

## Figures and Tables

**Figure 1 ijms-23-15859-f001:**
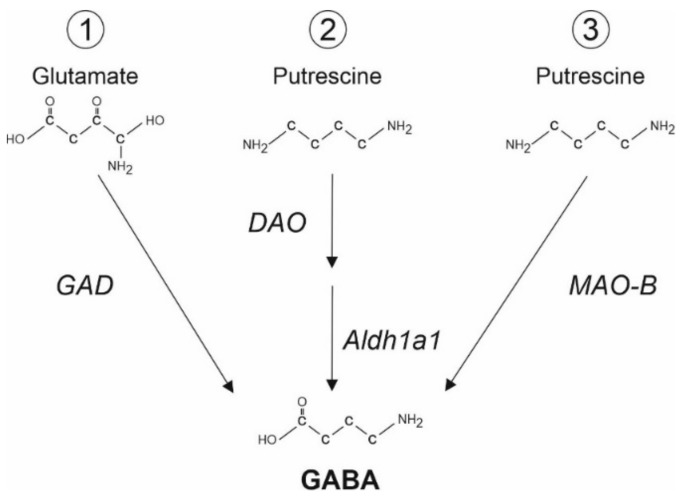
Several pathways of GABA synthesis in astrocytes. (**1**) Glutamate decarboxylase (GAD) can convert glutamate into GABA [28]. (**2**,**3**) GABA can be synthesized from putrescine, a monoamine, via (**2**) Diamine Oxydase (DAO) and Aldehyde Dehydrogenase 1a1 (Aldh1a1, [34]) or (**3**) Monoamine oxidase B (MAO-B, [35]).

**Figure 2 ijms-23-15859-f002:**
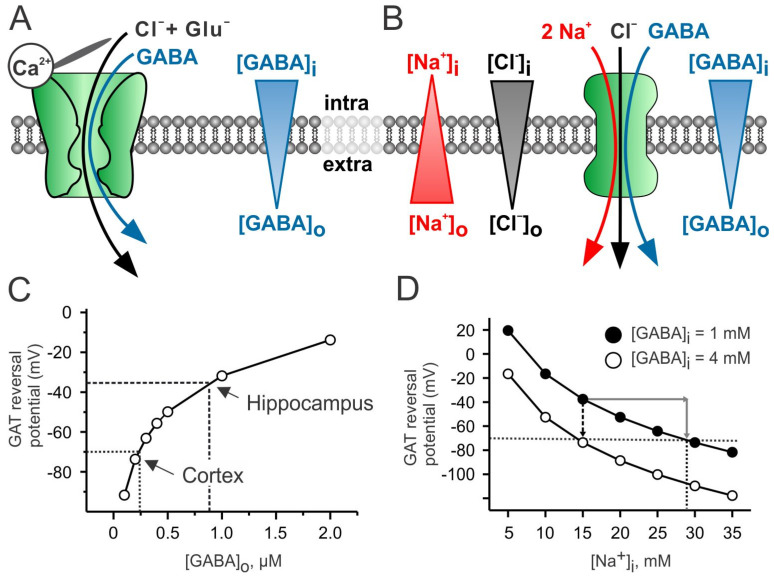
Possible pathways of GABA release from astrocytes. (**A**) GABA can be released via a Ca^2+^-activated anion channel (for example Bestrophin 1, Best1, [23]). The driving force for GABA release is the transmembrane GABA gradient. Best1 opening and conductance is controlled by [Ca^2+^]_i_. Note that Best1 is also permeable for Cl^−^ and glutamate. (**B**) GABA transporter (GAT3) operating in reverse mode can release GABA [20]. In addition to the GABA gradient, the transmembrane gradients of Na^+^ (quadratic impact) and Cl^−^ determine the electromotive driving force and, in turn, the direction of GABA transport. (**C**) The GAT reversal potentials depend on the [GABA]_o_. To calculate GAT reversal potential, following values have been taken: [Na^+^]_o_ = 140 mM, [Cl^−^]_i_ = 135 mM, [Na^+^]_i_ = 15 mM, [Cl^−^]_i_ = 40 mM, and [GABA]_i_ = 1 mM. The reported [GABA]_o_ of 0.2 µM in the cortex [40] and 0.8 µM in the hippocampus [41] favor the GAT3-mediated release under resting conditions in the former (GAT_rev_ ~ −70 mV) but not in the latter (GAT_rev_ ~ −35 mV). (**D**) To change the direction of GAT-mediated transport in the hippocampus, either [Na^+^]_i_ has to be increased to about 25–30 mM (gray arrow; for instance, as a result of EAAT-mediated glutamate uptake [21]) or [GABA]_i_, it has to be elevated to about 4 mM (dashed arrow) [28].

**Figure 3 ijms-23-15859-f003:**
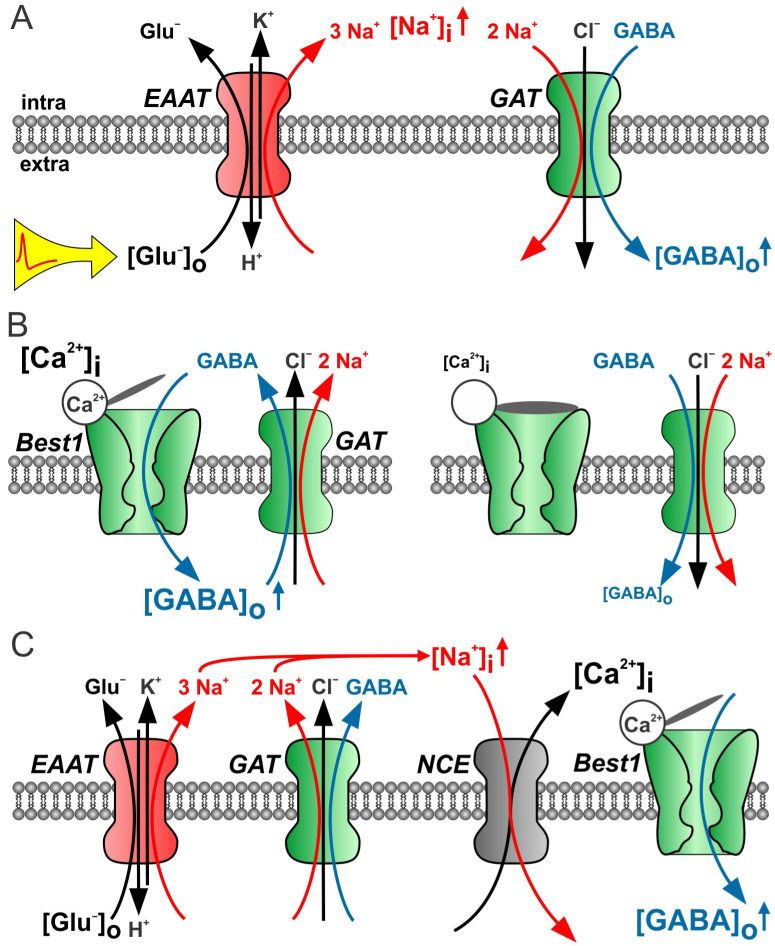
Best1- and/or GAT3-mediated GABA release from astrocytes. (**A**) Upon elevated extracellular glutamate concentrations, EAAT-mediated glutamate uptake increases [Na^+^]_I_ and switches GAT into reverse mode [21] (**B**) High [Ca^2+^]_i_ opens Best1 channels. Elevated [GABA]_o_ supports the uptake mode of GATs. Lowering of [Ca^2+^]_i_ or Best1 blockade results in [GABA]_o_ reduction and switches GAT into reverse mode (indirect in [34]). (**C**) Glutamate (EAAT) and/or GABA (GAT) uptake switch Na^+^-Ca^2+^ exchange (NCE) in reverse mode. Local [Ca^2+^]_I_ increase opens Best1 and enables GABA release. Note that local [Ca^2+^]_I_ increase may also affect GAT trafficking [81].

**Table 1 ijms-23-15859-t001:** Best1-channel- and GAT3-mediated GABA release in mouse models.

Brain Region	Disease Model	Best1	GAT3	References
Cerebellum	WT	+		[23]
Thalamus	WT	+		[34]
Cerebral Cortex	WT		+	[20,76]
Striatum	WT		+	[101]
Hippocampus	5xFAD (AD)			[28]
Hippocampus	APP^NL−F/NL−F^ knock-in (AD)		+	[91]
Hippocampus	APP/PS1 (AD)	+		[35]
Hippocampus	APP/PS1 (AD)	+		[35]
Striatum	Z-Q175-KI and R2/6 (HD)		+ (reduced)	[101]
Hippocampus	Kainate model (epilepsy)		+	[94]
Hippocampus	WAG/Rij (epilepsy)		+	[118]

## Data Availability

Not applicable.

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
