# Peer review of "GABA Release from Astrocytes in Health and Disease"

_ijms, 2022, doi:10.3390/ijms232415859_

Round 1
Reviewer 1 Report
Werner Kilb and Sergei Kirischuk wrote a very interesting review about the release of GABA from astrocytes, considering that most of the scientific community is focused on the release of GABA from neurons. They have argued and fine-tuned the brain-specific role of both BEST1 and GAT3 in the GABA release and uptake under both physiological and pathological conditions. However, since most of the studies on the Bestrophins have been concentrated on its expression and function in the retina and its role on neuroinflammation, it could also be important to describe this aspect and how an impaired GABA release can affect the onset and progression of retinopathies. Importantly, in astrocytes, both the glutamate transporter GLT1 and GABA transporter GAT-3 are critical for regulating the excitation/inhibition balance in pathological conditions as Tourette syndrome and autism spectrum disorders. I suggest to add a section describing the Best1-/GAT3 involvement in GABA release from astrocytes in autism spectrum disorders. Finally, I think it is also important to mention the crosstalk between astrocytes and neurons in the release of GABA, especially in the section 5 in which the mechanisms of interaction between BEST1 and GAT3 are hypothesized.
There are some typing errors in the lines 269 and 315.
Overall I believe that this review is well written, interesting and describes a new topic related to the release of gaba by astrocytes in the CNS, providing an important contribution to the mechanisms related to the release and uptake of gaba synthesized not only from glutamate but also from putrescine.
Author Response
Reviewer #1:
Werner Kilb and Sergei Kirischuk wrote a very interesting review about the release of GABA from astrocytes, considering that most of the scientific community is focused on the release of GABA from neurons. They have argued and fine-tuned the brain-specific role of both BEST1 and GAT3 in the GABA release and uptake under both physiological and pathological conditions.
…..
Overall I believe that this review is well written, interesting and describes a new topic related to the release of gaba by astrocytes in the CNS, providing an important contribution to the mechanisms related to the release and uptake of gaba synthesized not only from glutamate but also from putrescine.
We thank the reviewer for this kind evaluation.
Point 1) However, since most of the studies on the Bestrophins have been concentrated on its expression and function in the retina and its role on neuroinflammation, it could also be important to describe this aspect and how an impaired GABA release can affect the onset and progression of retinopathies.
In order to follow the reviewer’s comment, we added a short para on this topic in lines 115-127.
Point 2) Importantly, in astrocytes, both the glutamate transporter GLT1 and GABA transporter GAT-3 are critical for regulating the excitation/inhibition balance in pathological conditions as Tourette syndrome and autism spectrum disorders. I suggest to add a section describing the Best1-/GAT3 involvement in GABA release from astrocytes in autism spectrum disorders.
In order to follow the suggestion of the reviewer, we discuss the contribution of Best1 and GAT3 in autism spectrum disorders in chapter 4.3 (lines 372-399).
3) Finally, I think it is also important to mention the crosstalk between astrocytes and neurons in the release of GABA, especially in the section 5 in which the mechanisms of interaction between BEST1 and GAT3 are hypothesized.
In order to follow the reviewer’s proposal, we created a new section 5.4, in which we discussed how neuronal activity can modulate GAT3/Best1 function by neuroglial interactions (line 532-544).
There are some typing errors in the lines 269 and 315.
We corrected these typos.
Reviewer 2 Report
I have received the review article entitled"GABA release from astrocytes in health and disease: Bestrophin1-channel- and/or GABA transporter 3-mediated" by Werner Kilb and Sergei Kirischuk for evaluation.
In this manuscript, authors have reviewed the previous literature describing the Bestrophin1-channel- and/or GABA transporter 3-dependent GABA release in various pathological diseases. the review is well written and updated but I have some comments to improve the quality of this review. My comments are given below:
Major comment:
1- The topic of review is very important and timely however I would suggest to authors to add the graphical abstract to make it more attractive and readable.
2- Introduction section need to be rewrite because many facts are missing in this including introduction of Bestrophin1-channel and its role in various neurological diseases.
3- I would suggest authors to add a table showing facts about the involvement of astrocytes mediated GABA release, Bestrophin1-channel- and/or GABA transporter 3 and neurological diseases in human as well as in animal models.
Minor comments:
1- The title of the review should be re-phrased.
2-In introduction section, Line no. 43 in the proper citation?
3- Quality and clarity of Figure 2 should be improved.
Author Response
Reviewer #2:
I have received the review article entitled "GABA release from astrocytes in health and disease: Bestrophin1-channel- and/or GABA transporter 3-mediated" by Werner Kilb and Sergei Kirischuk for evaluation. In this manuscript, authors have reviewed the previous literature describing the Bestrophin1-channel- and/or GABA transporter 3-dependent GABA release in various pathological diseases. the review is well written and updated but I have some comments to improve the quality of this review. My comments are given below:
We thank the reviewer for this kind evaluation.
Major comment 1: The topic of review is very important and timely however I would suggest to authors to add the graphical abstract to make it more attractive and readable.
In order to follow the reviewer’s proposal, we draw a graphical abstract representing the two major pathways for astrocytic GABA release. Our attempts to include also the biochemical details of GABA synthesis in this scheme resulted in a non-convincing, overloaded figure. Therefore we decided to use only this version with a condensed information.
Major comment 2: Introduction section need to be rewrite because many facts are missing in this including introduction of Bestrophin1-channel and its role in various neurological diseases.
In order to follow the reviewer’s concern we rewrote the introduction, now giving more emphasize to Bestorphin1 and its role in neurological diseases (lines 37-52 and lines 115-127). See also our response to a comparable suggestion by reviewer #1.
Major comment 3: I would suggest authors to add a table showing facts about the involvement of astrocytes mediated GABA release, Bestrophin1-channel- and/or GABA transporter 3 and neurological diseases in human as well as in animal models.
In order to follow the reviewer’s suggestion, we added a table (Table 1, line 465) summarizing the contribution of Best1 and GAT3 under physiological conditions and for mouse models of several diseases.
We also considered include human pathophysiological conditions into this table, but unfortunately in most reports on human disorders only the changes in GABA or GAT/Best levels have been reported without mentioning the release mechanisms. Therefore, we have listed in the Table only data obtained in mouse models.
Minor comment 1: The title of the review should be re-phrased.
We rephrased the title to “GABA release from astrocytes in health and disease”.
Minor comment 2. In introduction section, Line no. 43 in the proper citation?
In order to follow the reviewer’s concern, we added better references for this statement (now line 57).
Minor comment 3: Quality and clarity of Figure 2 should be improved.
In order to follow the reviewer’s concern, we redraw the Figure 2. In this respect, we also altered Figure 3, to have similar symbols for membrane transporters throughout the manuscript.
Round 2
Reviewer 2 Report
Accepted